# Survey of MRI Usefulness for the Clinical Assessment of Bone Microstructure

**DOI:** 10.3390/ijms22052509

**Published:** 2021-03-02

**Authors:** Enrico Soldati, Francesca Rossi, Jerome Vicente, Daphne Guenoun, Martine Pithioux, Stefano Iotti, Emil Malucelli, David Bendahan

**Affiliations:** 1CRMBM, CNRS, Aix Marseille University, 13385 Marseille, France; david.bendahan@univ-amu.fr; 2IUSTI, CNRS, Aix Marseille University, 13013 Marseille, France; jerome.vicente@univ-amu.fr; 3ISM, CNRS, Aix Marseille University, 13288 Marseille, France; daphne.guenoun@ap-hm.fr (D.G.); martine.pithioux@univ-amu.fr (M.P.); 4Department of Pharmacy and Biotechnology, University of Bologna, 40126 Bologna, Italy; francesca.rossi105@unibo.it (F.R.); stefano.iotti@unibo.it (S.I.); emil.malucelli@unibo.it (E.M.); 5Department of Radiology, Institute for Locomotion, Saint-Marguerite Hospital, ISM, CNRS, APHM, Aix Marseille University, 13274 Marseille, France; 6Department of Orthopedics and Traumatology, Institute for Locomotion, Saint-Marguerite Hospital, ISM, CNRS, APHM, Aix Marseille University, 13274 Marseille, France; 7National Institute of Biostructures and Biosystems, 00136 Rome, Italy

**Keywords:** MRI, bone microarchitecture, bone morphology, bone quality

## Abstract

Bone microarchitecture has been shown to provide useful information regarding the evaluation of skeleton quality with an added value to areal bone mineral density, which can be used for the diagnosis of several bone diseases. Bone mineral density estimated from dual-energy X-ray absorptiometry (DXA) has shown to be a limited tool to identify patients’ risk stratification and therapy delivery. Magnetic resonance imaging (MRI) has been proposed as another technique to assess bone quality and fracture risk by evaluating the bone structure and microarchitecture. To date, MRI is the only completely non-invasive and non-ionizing imaging modality that can assess both cortical and trabecular bone in vivo. In this review article, we reported a survey regarding the clinically relevant information MRI could provide for the assessment of the inner trabecular morphology of different bone segments. The last section will be devoted to the upcoming MRI applications (MR spectroscopy and chemical shift encoding MRI, solid state MRI and quantitative susceptibility mapping), which could provide additional biomarkers for the assessment of bone microarchitecture.

## 1. Introduction

### 1.1. Bone Disorders and Investigative Tools

A large number of studies have demonstrated the substantial burden of bone disorders worldwide [1,2,3]. Considered as the second greatest cause of disability [1], musculoskeletal pathologies account for 6.8% of total disability worldwide [2]. Bone pathologies are usually affecting the bones solid phase, which is composed of both cortical and cancellous/trabecular types of bone. Bone alterations commonly include cortical shell thinning, increased porosity of both cortical and trabecular bone phases [4,5], and reduced density, volume, and regenerative power. These bone modifications generally account for a reduced resistivity and flexibility eventually leading to an increased risk of fragility fractures accompanied by long-term disabilities. Recent studies have shown that people over the age of 50 with a high risk of osteoporotic fractures represented more than 150 million people worldwide with 137 million women [6]. This number is expected to exceed 300 million by 2040 [6]. Fragility fractures lead to more than half a million hospitalizations each year in North America alone, with an annual direct cost, which has been estimated to be $17 billion dollars in 2005. This cost is expected to rise by almost 50% by 2025 [7]. Overall, the early identification of bone fragility risk is a major health issue [8]. In the clinical context, bone disorders are usually assessed using dual-energy X-ray absorptiometry (DXA), which is able to assess the bone mineral density (BMD). The BMD score is then compared to a reference range of values calculated in healthy (25–35 years old) volunteers taking into account sex and ethnicity. Accordingly, a score (T-score) is generated indicating how far, in terms of SD (standard deviation), the measured BMD is from the reference values. A T-score between −1 and −2.5 indicates a low bone mass or osteopenia while a value lower than −2.5 is indicative of osteoporosis. The corresponding method has good sensitivity (around 88% for both men and post-menopausal women), but the specificity is poor (around 41% for post-menopausal women and 55% for men) [9] resulting in a low clinical diagnostic accuracy (70%) [10]. In addition, DXA measurements do not take into consideration microarchitectural alterations, which have also been recognized as part of the structural picture in osteoporosis. Of interest, bone microarchitecture can be assessed using quantitative computed tomography (qCT) [11,12]. Given that both DXA and qCT are both radiative imaging techniques, non-radiative alternatives would be of great interest. Over the last decades, magnetic resonance imaging (MRI) [13,14,15] has been indicated as a non-ionizing and non-invasive technique.

Using MRI, a large number of studies have attempted to assess bone microarchitecture in bone disorders and more particularly in osteoporosis [16,17,18,19]. The corresponding studies have been conducted at different magnetic field strengths, using different Radio Frequency coils and pulse sequences. Although, the results were compelling, the sensitivity of the corresponding microarchitecture metrics for diagnostic purposes and the assessment of the disease severity is still a matter of debate.

On the basis of a comparative survey of MRI, computed tomography, and DXA-based metrics, we intended to address the issues related to the diagnostic potential of the corresponding metrics and their capacity to predict disease severity. The final section will be devoted to potential perspectives offered by magnetic resonance spectroscopy (MRS) and chemical shift encoding (CSE-MRI), solid-state MRI, and quantitative susceptibility mapping (QSM).

### 1.2. Bone Microstructure

Bone is a multiphase material composed of a solid phase and a viscoelastic component. The solid phase is considered as hierarchical, anisotropic, and heterogeneous and is composed of 65% of inorganic matrix (mostly calcium hydroxyapatite crystals) and 35% of organic matrix (type I collagen, proteoglycans, and bound water) [20]. While the inorganic matrix is characterized by a high rigidity, a high resistivity, and an elastic behavior, the organic matrix is deformable thereby providing the tissue with tensile strength. Due to the combination of these two materials, bone tissue is simultaneously deformable and rigid [21]. The solid phase creates a shell for the bone marrow, which is the viscoelastic component. The bone marrow on the other hand has a double function. It provides nutriments to the solid phase allowing higher regenerative rate and is able, due to its viscoelastic properties, to spread the dynamics of an impulsive action, reducing the risk of fractures due to impacts [22]. Bone tissue is composed of both trabecular and cortical bone phases. Cortical bone covers the whole surface of the bone. It is compact, dense, and characterized by overlapped and parallel lamellae, which provide a large resistivity [20]. Trabecular bone is the inner compartment of bone tissue. It is composed of 25% of bone and 75% of marrow [23]. At the microstructural level, trabecular bone appears as a complex 3D network of interconnected trabeculae rods and plates responsible for tissue resistance to loading forces. The bone inner architecture is an important contributor to bone strength independent of bone mass [20]. It is characterized by a high porosity so that trabecular bone is lighter and less dense than cortical bone. In fact, cortical bone mainly works in compression while trabecular bone principally works in flexion and torsion reaching a higher area under the stress–strain curve [23].

Bone is actually a dynamic porous structure and this porosity can change as a result of pathological processes but also as an adaptive response to mechanical or physiological stimuli. This change in both cortical and trabecular bone porosity can strongly affect the corresponding mechanical properties [23].

## 2. Bone Pathologies and Clinical Approach

### 2.1. Principal Bone Pathologies

Pathologies of the bone microstructure are quite common. Musculoskeletal (MSK) complaints are the second most common reason for consulting a medical doctor and account for 10–20% of primary care visits [4]. They are also among the leading causes of long-term disability and the leading cause for long-term absence from work in numerous countries [4,24]. Worldwide, the total number of MSK disability adjusted life years (DALYs) significantly increased from 80.2 million in 2000 to 107.9 million in 2015 (*p* < 0.001) with the total number of MSK years lived with disability (YLDs) increasing from 77.4 million to 103.8 million. Overall, MSK diseases represent the second cause of YLDs worldwide [25].

The most common pathology related to bone microstructure alterations is osteoporosis in which bone density and volume of bone segments or specific bone regions can be progressively reduced. Patients with osteoporosis are at high risk of having one or more fragility fractures, which eventually lead to a physical debilitation and potentially to a downward spiral in physical and mental health. Johnell et al. reported that 9 million osteoporotic fractures occurred in 2000, 1.6 million in the hip, 1.7 million in the wrist, and 1.4 million in the vertebrae [5]. Only in the five largest countries in Europe plus Sweden (EU6), the number of fragility fractures were estimated at 2.7 million in 2017 with an associated annual cost of €37.5 billion and both fragility fractures and associated annual cost are expected to increase by 23% in 2030 [26]. A large Chinese epidemiological survey using DXA among people aged fifty years or older demonstrated the prevalence of osteoporosis in males (6.46%) than females (29.13%) meaning that there are 10 million men and 40 million women with osteoporosis only in China [27]. The current DALYs per 100 individuals age 50 years or more were estimated at 21 years, which is higher than the estimates for stroke or chronic obstructive pulmonary disease [26]. Moreover, it has been reported that if all patients who fracture in the EU6 were enrolled into fracture liaison services, at least 19,000 fractures every year might be avoided [26]. Finally, Kemmak et al. reported that the direct annual cost of treating osteoporotic fractures of people on average is between 5000 and 6500 billion USD in Canada, Europe, and USA alone, not taking into account indirect costs, i.e., disability and loss of productivity [28].

Osteoporosis may be linked to ageing, particularly in postmenopausal women, or can occur as a result of specific conditions, i.e., diabetes, anorexia nervosa, and obesity or treatments, i.e., corticosteroid. Indeed, corticosteroid-induced osteoporosis is the most common form of secondary osteoporosis and the first cause in young people. Bone loss occurs early after the initiation of corticosteroid therapy and is correlated to dose and treatment duration [29]. Fragility fractures have been associated with early mortality and increased morbidity having a significant effect on the quality of life of both patients affected by diabetes [30,31,32], anorexia nervosa [33,34], and obesity [35,36,37].

Additionally, a number of childhood diseases cause rickets, a physical condition resulting from a delayed calcium phosphate mineral deposition in growing bones, which may lead to skeletal deformities [38]. In adults, the equivalent disease is called osteomalacia and may have devastating consequences if not diagnosed and treated [39,40]. Patients with chronic renal disease are at risk of developing a complex bone disease known as renal osteodystrophy, which is responsible for an increasing bone resorption due to an increased osteoclast activity [11,41].

Paget’s disease is a chronic progressive bone disorder occurring in middle-aged or older adults and which commonly affect spine, pelvis, legs, or skull [42,43]. The most likely etiology is a slow paramyxoviral viral infection in generally susceptible individuals, however the exact cause is unknown [44]. It appears to arise more or less simultaneously in one or more skeletal sites, remaining restricted there. In long bones the disease first appears in the region of the proximal epiphysis and advances along the shaft at a rate of 8 mm/yr [45]. Paget’s disease is initially characterized by bone resorption, disorganized bone deposition, resulting in pathological bone remodeling where the osteoclastic activity is predominant, is then followed by a mixed phase of osteoclast and osteoblast with osteoblasts prevailing, and end with an inactive phase where the osteoblastic activity declines [42]. The leading edge of this advance is often visible as a V-shaped “lytic wedge” reflecting osteoclastic resorption [45]. Moreover, it has been noticed that an elevated serum alkaline phosphatase level correlated with the disease activity [44]. Diagnosis and follow-up are usually based on imaging modalities in order to assess disease status, bone microarchitecture and metabolic activity. MRI is invaluable for the assessment of complications, i.e., spinal stenosis and sarcomatous degeneration [42]. An early diagnosis of this disease can minimize the impact on the patient quality of life [46].

Many genetic and developmental disorders can affect the skeleton. Among them, the most common is the osteogenesis imperfecta [47]. Osteogenesis Imperfecta (OI) is a systemic connective tissue disorder characterized by low bone mass and bone fragility causing significant morbidity due to pain, immobility, skeletal deformities, and growth deficiency. Moreover, it is the most prevalent heritable bone fragility disorder in children [47,48,49,50,51]. OI is a skeletal dysplasia characterized by bone fragility and high incidence of fractures that may occur with minimal or no trauma [48,50]. Fractures may involve atypical locations. Vertebral fractures occur in about 70% of OI patients. Joint hypermobility is also common and gray or blue scleral hue is a predominant OI feature [48,50]. Moreover, severe OI may present prenatally by detection of in utero fractures and shortening of long bones on prenatal ultrasound [47,48,50]. OI is currently diagnosed using patient history, clinical examination, lumbar spine BMD, bone biochemistry, and image analysis (CT and/or MRI scans) [47,50,52]. Interestingly, Ashinsky et al. have shown that multiparametric classification using quantitative MRI could detect at the skin level differences between OI patients and unaffected individuals, suggesting the potential of MRI for the clinical OI diagnosis [53]. However, the molecular diagnosis using DNA sequence analysis can pinpoint the exact OI cause and provide information about the recurrence risk to affected individuals and their families [47,49]. There is no cure for OI and among the clinical and chirurgical therapies (largely supportive at present), the bisphosphonate therapy has shown remarkable effect where treatment efficacy and follow ups are usually assessed though image analysis [47,52,54].

Lastly, some skeletal disorders can result from primary or secondary tumors. Primary bone tumors are rare, accounting for < 0.2% of malignant neoplasms registered in the EUROCARE (European Cancer Registry based study on survival and care of cancer patients) database [55], and in particular osteosarcoma (OS) represents < 1% of all cancers diagnosed in the United States [56]. OS is classically described as a high grade spindle shaped neoplasm with malignant cells that produce osteoid [56]. However, OS is the first primary cancer of bone (incidence: 0.3 per 100,000 per year) with a relatively high incidence in the second decade of life (incidence: 0.8–1.1 per 100,000 per year at age 15–19 years) [55]. Most of the OSs of younger patients arise in the metaphysis of long bones with the most common sites being the extremities (distal femur, proximal tibia, and proximal humerus) [56], while the axial tumor sites increases with age. Conventional radiography is the first radiological investigation. However, MRI investigation of the whole compartment with adjacent joints is regarded today as the best modality for local staging of extremities and pelvic tumors [55,57]. The final diagnosis as for the disease grading is based on biopsy, histology, and molecular assessment. Curative treatment consists of chemotherapy and surgery [55,57]. In the case of chemotherapy treatment, dynamic MRI is reliable for the evaluation of changes in tumor vascularity [57,58].

### 2.2. Clinical Approach

The clinical evaluation of bone status is mainly based on the dual-energy X-ray absorptiometry (DXA), which gives information about the bone mineral density (BMD). The whole body or a bone segment is scanned using X-rays and a 2D projection of bone density is evaluated using a standard reference. Although this technique has been classified as being minimally invasive, the radiation deposition dose for a whole body DXA examination is 0.0042 mSv, and can reach up to 0.009 mSv and 0.013 mSv respectively for the hip and the spine examination [59]. DXA is the most common screening evaluation test for osteoporosis and body composition (whole body percent fat). Since the proportion of cortical bone is often larger, DXA is more sensitive to the presence and quality of the cortical bone. DXA can then be considered as poorly sensitive to trabecular bone alterations. Accordingly, recent studies have shown that DXA is not well suited to discriminate between patients with and without fragility fractures while this would be possible with quantitative microarchitecture analysis [18,60,61,62].

For clinical applications, bone inner morphology can be assessed using quantitative computed tomography (qCT), an X-ray based technique, which could be used to assess central and peripheral skeletal sites. The acquired volumes of interests are reconstructed from a stack of images, which can reach *n* = 900 images for a 45-cm abdominal-thorax scan, with a radiation deposition dose of 0.06–0.3 mSv/image. On that basis, this technique is considered as highly radiative [59,63]. On the contrary to DXA, the tomographic reconstruction and the resolution power of qCT can provide information related to the inner bone morphology. For deep bone segments, the corresponding image resolution ranges between 0.160 and 0.300 mm due to radiations issues [64]. For bone extremities, i.e., radii and tibiae, qCT can be replaced by high-resolution peripheral computed tomography (HR-pQCT), which provides a higher resolution, i.e., between 0.040 and 0.150 mm [12,64]. Due to a low benefit–risk ratio, qCT and HR-pQCT are not currently used for the diagnosis of bone diseases in clinical practice.

## 3. MRI Based Approach

A non-invasive alternative to DXA and qCT could be MRI. Over the last two decades, a large number of studies have intended to assess bone microstructure using MRI. The initial investigations have been performed using T1-weighted spin echo sequences characterized by short TR (<1200 ms) and short TE (<25 ms) in distal radius and calcaneus [16,65,66,67]. Due to technical advances, tibiae [17,68,69], spine [65,70], and proximal femur [18,60,71,72] have been investigated. MRI of trabecular microstructure can be obtained by imaging the marrow phase inside the bone segment, which appears as a hyperintense signal in conventional MR images. Using higher field MRI, i.e., 3T one can expect an increased signal to noise ratio (SNR), which can be translated either in a reduced acquisition time or an increased image resolution. Over the last decades, due to the higher availability of high-field (HF) MRI scanners, a large number of studies have been dedicated to the MRI assessment of osteoporosis [17,18,60,62,69,71,72]. Very recently, clinical FDA and CE-approved ultra-high field (i.e., 7T UHF) MRI scanners with announced MSK applications have become available. Their clinical availability is still poor and the coming results will be of utmost importance to decide about the future of UHF MRI for clinical purposes.

Using MRI, the most common extrapolated features are the bone volume fraction (BVF), the trabecular thickness (Tb.Th), spacing (Tb.Sp), and number (Tb.N) [18,62].

### 3.1. Technical Considerations for Clinical Usefulness

A signal to noise ratio (SNR) of 10 has been reported as the minimum value for the investigation of bone microarchitecture [73]. The scan time considered acceptable for clinical examination has to range between 10 and 15 min. As a result the minimum voxel size, which has been obtained at 1.5T was between 0.135 and 0.250 mm while the slice thickness was between 0.3 and 1.5 mm. One has to keep in mind that SNR would be higher for superficial anatomical sites (radius or calcaneus compared to deeper anatomical sites, e.g., proximal femur) leading to higher resolution or shorter acquisition time. Moreover, SNR can be increased at higher field strengths and/or using multichannel coils [73,74,75,76].

MRI pulse sequences such as gradient recalled echo (GRE) and spin echo (SE) have also been tested at different field strengths [17,71,77]. It has been shown that SE sequences were less susceptible to partial volume effects as compared to GRE sequences and that GRE were more sensitive to trabecular broadening than SE. These results indicate that SE sequences would provide more accurate results regarding trabecular characteristics [17,77]. However, the use of these pulse sequences might be problematic using ultra-high field (UHF) MRI considering power-deposition issues.

A list of the main literature references, scanned regions, sequences, and principal MRI setup parameters is reported in Table 1.

### 3.2. Microstructure Investigation

In the majority of MRI literature, the morphological parameters that are reported are BVF, Tb.Th, Tb.Sp, and Tb.N [71,77,78]. In addition, some groups have proposed some other features such as an erosion index, trabecular rod- and plate-like structures, trabecular plate-to-rod ratio, trabecular isolation, and fractal lacunarity [18,79].

These microarchitectural parameters have been generated from the post-processing of both 2D and 3D images. The corresponding analyses were performed in binarized images or in original grey level intensities. All these approaches have tried to take into account partial volume effects occurring given the poor resolution of MRI as compared to the trabeculae dimension [62,80,81]. So far, no standard reference has been suggested.

Studies performed at different MRI field strength in postmenopausal woman with fragility fractures have illustrated microstructural alterations (reduced BVF and increased Tb.Sp) whereas DXA T-scores were unchanged. In a study conducted in distal radii at 1.5T, Kijowsky et al. showed that post-menopausal woman had a slightly lower (−9%) bone volume fraction and a higher erosion index (+17%) compared to controls [82]. Krug et al. in a study conducted on the proximal femurs of six healthy males and females using both 1.5T and 3T MRI showed good correlation (r up to 0.86) between structural parameters obtained from the two different field strengths. However, they reported that bone structure of the proximal femur was substantially better depicted at 3T than 1.5T [71]. Microstructure alterations have been reported in a large variety of cases including chronic kidney disease (CKD) [11,41], HIV-infection [83] glucocorticoid-induced osteoporosis [72], or disuse osteoporosis [84].

In a 3T MRI study conducted in distal tibiae of 20 patients with CKD, Ruderman et al. reported trabecular deterioration together with reduced cortical thickness [41]. Moreover, a study conducted on 30 patients affected by end stage renal disease (ESRD) it has been shown that Tb.N, Tb.Th, and whole bone stiffness were significantly lower (*p* < 0.01) is ESRD compared to controls [85]. A similar study conducted on distal tibiae of 11 kidney transplant recipient patients have high-lightened post-transplant deterioration in trabecular bone quality [86]. In a study conducted in proximal femurs at 3T, glucocorticoid treated patients had a largely reduced (−50.3%) Tb.N, trabecular plate-to-rod ratio (−20.1%), and a largely increased (+191%) Tb.Sp [72]. Patients with a disuse osteoporosis displayed similar anomalies for BVF (−30%), Tb.N (−21%), Tb.Th (−12%), and Tb.Sp (+48%) [84]. Chang et al. [18] further supported and extended these results in a study conducted in distal femur at 7T. In 31 subjects with fragility fractures, they reported a lower BVF (–3%), Tb.N (–6%), and erosion index (–6%). Moreover, in a 7T MRI study conducted in the distal radius of 24 women, Griffin et al. reported a trabecular bone microarchitecture gradient with an overall higher quality (+123% BVF, +16% Tb.N) distally (epiphysis) than proximally (diaphysis) [87].

Ultra-high field MRI can provide images with a smaller pixel size (0.156 mm × 0.156 mm) as compared to the resolution achieved at lower field strength (0.234 mm × 0.234 mm for example at 3T [17,71]). In a dual 3T-7T study conducted in distal tibiae of 10 healthy volunteers, Krug et al. reported that metrics computed at higher field strength were different than those quantified from 3T MR images. More specifically, UHF measurements illustrated increased BVF (+22%) and Tb.Th (+25%) whereas Tb.Sp (−21%) and Tb.N (−4%) were both decreased [88]. These results suggest a higher discriminative power of UHF MRI for trabecular features.

### 3.3. Microstructure vs. DXA

In a study conducted in 32 postmenopausal women, Kang et al. showed a good correlation between DXA-based BMD and MRI T2 and T2 * in calcaneus (r = −0.8, *p* < 0.001) and spine (r = −0.53, *p* = 0.002) [89]. Similar results have been reported for the femoral neck [65,90] with a good correlation (r = 0.74, *p* < 0.001) between DXA-based BMD and T2 * values [91]. T2 * relaxation time illustrates the susceptibility differences between trabecular and bone marrow leading to signal loss due to magnetic field inhomogeneities. MRI-derived T2 * has been shown to correlate with DXA results in several anatomical areas such as calcaneus, distal radius, and Ward’s area in the femoral neck [92,93]. Based on T2 * measurements, Schmeel et al. reported a significant difference between benign and malignant neoplastic vertebral compression fractures (VCFs). A 72% diagnostic accuracy was computed [94]. Furthermore, a strong negative correlation was found between the pelvic bone marrow adipose tissue (BMAT) calculated in 56 healthy women using MRI and the corresponding DXA-based BMD (r = −0.646, *p* < 0.001) [95]. The negative correlation indicates that patients with decreased bone mineral density are characterized by an increased fat content in bone marrow [95,96,97].

Based on highly resolved MR images (0.150–0.300 mm in-plane pixel size), Chang et al. showed a lack of significant correlation between DXA-computed BMD T-scores and MRI computed microarchitectural parameters in the femoral neck in both controls and glucocorticoid-treated patients [72,98]. Similar results were also reported more recently in subchondral tibiae [61], proximal femurs [62,81], vertebrae [99], and on patients affected by diabetes [31,32]. Guenoun et al. reported that the combination of BVF and BMD was able to improve the prediction of the failure stress (from r^2^ = 0.384 for BMD alone to r^2^ = 0.414). All the presented results suggest that although density and structure metrics illustrate bone quality, microarchitectural parameters provide additional information regarding skeletal fragility.

### 3.4. Voxel Size and Microstructure

Results from the literature showed that image resolution is a key parameter for the assessment of bone microarchitecture. Importantly, a distinction must be made between in-plane and through-planes resolution. For specific oriented plane (mostly perpendicular to the trabecular), an in-plane MRI pixel size in the same order of magnitude than Tb.Th dimension is enough to measure morphological parameters similar to those extrapolated using gold standard method and so both ex vivo (µCT) [81,100] and in vivo (HR-pQCT) [77]. If one intends to assess bone microstructure using small isovolumetric voxels (0.15 mm), close to the actual thickness of the trabeculae, with an acceptable SNR, acquisition times would exceed the in vivo acceptable duration. One can increase the SNR and reduce the acquisition time with an increased slice thickness while keeping the plane pixel size constant. Accordingly, the radius morphological parameters computed from similar in-plane pixel sizes and different slice thicknesses (0.156 mm × 0.156 mm × 0.3 mm [13], 0.156 mm × 0.156 mm × 0.5 mm [16,101], 0.156 mm × 0.156 mm × 0.7 mm [65], and 0.153 mm × 0.153 mm × 0.9 mm [102]) were comparable. In fact, the bone inner microarchitecture appeared to be a mixture of oriented plates- and rod-like structures. The parallel trabecular plates structures are separated by bone marrow and are perpendicular to the coronal plane [103]. On that basis, increasing the in-plane pixel size should provide more accurate results independently of the slice thickness. As reported by Mulder et al., the calculated volume of ellipsoid at high resolution (0.1 mm × 0.1 mm) is independent from the anisotropy factor but related to the orientation [104].

Different studies performed in distal radii at 1.5T, using similar in-plane pixel size and using different slice thicknesses above 0.3 mm, reported comparable morphological results [13,16,82]. However, in a study conducted by Majumdar et al. in 39 distal radii specimens acquired using 1.5T MRI and contact radiograph, 0.9-mm thick MR images performed better than those obtained from 0.3-mm images. This was explained with the significantly higher SNR (18.2 in 0.9-mm thick images and 9.3 in 0.3-mm sections) [102]. Similar results were obtained in vivo in distal radii scanned at 1.5T (0.156 mm × 0.156 mm × 0.5 mm) with an acceptable SNR around 10 [75]. Moreover, wrists and distal tibiae scanned in patients using 1.5T with pixel sizes in the same range of trabecular thickness (0.156 mm × 0.156 mm × 0.410 mm [66] and 0.137 mm × 0.137 mm × 0.410 mm [105]) reporting lower acquisition time for wrist (12 min) than for tibiae (16 min) and good image quality in both anatomical regions. In a second study conducted by Majumdar et al., 31 cadaveric proximal femurs were scanned at 1.5T with an in-plane pixel size of 0.195 × 0.195 and comparing two different slice thicknesses (0.9 and 0.3 mm). The SNR achieved was 25.2 and 13.8 for the larger and smaller slice thickness respectively. The corresponding acquisition times were very long (27:19 and 73:14 min), i.e., much longer than what could be accepted in clinics [106].

The knee articulation has also been assessed in the study of Rajapakse et al., 17 distal tibiae specimens were scanned at 3T (0.137 mm × 0.137 mm × 0.410 mm) in 7 min [19]. These results where extended in vivo by Zhang et al., in the distal tibiae of 20 postmenopausal women with osteoporosis. The scanning time using 3T MRI (0.137 mm × 0.137 mm × 0.410 mm) was less than 15 min [69]. Krug et al. further confirmed these results in a study comparing 3T MRI (0.156 mm × 0.156 mm × 0.5 mm) and X-ray based techniques both ex vivo (5 tibiae and 3 radii) and in vivo (5 radii and 6 tibiae). While the scanning time was less than 10 min, correlations were reported between both methods and so for the whole set of parameters, i.e., BVF (r = 0.83) and Tb.Sp (r = 0.7) [107]. Liu et al. also reported 3T MR images (pixel size 0.180 mm × 0.180 mm, acq. time 9:18 min) of 92 distal femurs divided in three groups (without osteoarthritis, mild osteoarthritis, and severe osteoarthritis) reporting progressively lower BVF and higher erosion index from healthy patients to those affected by severe osteoarthritis [108], extending previous results [109,110,111].

### 3.5. Main Magnetic Field Strength Effect

The technical advantages of moving from 1.5T to 3T or 7T MR scanners were clearly visible in the acquisition of deeper anatomical sites keeping the spatial pixel size in the same order of the trabecular thickness, the acquisition time (acq. Time), and the SNR (>10) being clinically compatible. On that basis, 7T MR scanners have been tested mostly for the acquisition of distal and proximal femur, which represent a clinical important fracture site and one of the most invalidating [8].

In a comparative study conducted in vivo in proximal femur at 1.5 and 3T, Krug et al., reported as expected a 1.6 time-SNR increase together with a corresponding contrast-to-noise ratio (CNR) increase at higher magnetic field. While the 3T images clearly showed the trabecular bone structure, the image resolution did not allow a proper trabecular morphological analysis [71]. In a more recent study in the knee joint of 16 healthy volunteers scanned at 1.5T (0.6 mm × 0.6 mm × 0.6 mm, acq. time 7:15 min) and 3T (0.5 mm × 0.5 mm × 0.5 mm, acq. time 6:51 min), Abdulaal et al. reported significantly higher SNR (*p* < 0.05) allowing a better trabecular characterization at 3T than 1.5T [112]. Moreover, 3T MRI could be used to successfully scan radii with an in-plane pixel size comparable to the trabecular thickness and an acquisition time (10 min) lower than what commonly needed at 1.5T [101,113]. Jarraya et al., on a study conducted in 50 distal radii scanned at both 3T (0.2 mm × 0.2 mm × 2.0 mm, acq. time 4:29 min) and 7T MR (0.125 mm × 0.125 mm × 2.0 mm, acq. time 3:16 min), reported a statistical significant difference of horizontal and fractal dimensions between patients with chronic wrist disease and controls [114]. A similar comparative analysis has been performed between 3T and 7T MRI (0.156 mm × 0.156 mm × 0.5 mm, acq. time lower than 10 min) and HR-pQCT. Krug et al. showed that tibial trabecular structures were over-represented at higher field strength. Due to susceptibility-induced broadening smaller trabeculae normally not visible due to partial volume effects may be emphasized at 7T [88]. Moreover, using UHF MRI (0.234 mm × 0.234 mm × 1.0 mm, acq. time 7 min), Chang et al. reported that microarchitectural parameters could discriminate between patients and controls and could detect bone deterioration in women with fragility fractures for whom BMD was normal [18]. In addition to the effects of magnetic field strength, Krug et al. also assessed the potential differences between GRE and TSE sequences at 7T. SNR was slightly higher for GRE sequences (13.2 vs. 11.9) while the bone marrow signal was more homogeneous using TSE sequences. This large homogeneity is related to a reduced susceptibility-induced broadening of the trabeculae so that the morphological analysis showed decreased BVF (−13%) and Tb.Th (−23%). These values were closer to those reported using the HR-pQCT reference method [88]. Furthermore, in a study conducted in three cadaveric proximal femurs scanned at 7T (0.130 mm × 0.130 mm × 1.5 mm, acq. time 16 min) and using µCT, Soldati et al. reported no statistical difference between the methods and so for the whole set of morphological parameters [81]. These preliminary results strongly suggest that UHF MRI could be of interest for the in vivo assessment of bone microarchitecture particularly for the deep anatomical regions.

### 3.6. Comparison with CT Measurements

Validation of the bone morphological parameters derived from the high-resolution MR images has usually been performed through the comparison with X-ray based techniques (qCT, HRpQCT, and μCT).

#### 3.6.1. Ex-Vivo

Ex vivo studies have been performed in different body parts. However, due to the samples size (<5 cm^3^) and the commonly used preparation protocols (replacement of marrow), they remain poorly representative of the in vivo conditions [13,78,81,102,115]. One of the first studies validating MR bone structure measurements was performed by Hipp et al. in cubic bovine trabecular bone from several anatomical sites using optical and micro-MRI methods. BVF and Tb.N were linearly related (r^2^ = 0.81 and r^2^ = 0.53 respectively) and did not differ statistically (*p* = 0.96 and *p* = 0.17) [78,115]. These results were confirmed and extended in human specimens by Majumdar et al., in a study conducted in 7 cubic specimens of trabecular bone extracted from cadaveric radii scanned at 1.5T (0.156 mm × 0.156 mm × 0.3 mm) and using μCT (0.018 mm isovolumetric). The results showed a good correlation for the whole set of metrics with BVF and Tb.Th performing the best (r = 0.77 and 0.87 respectively) and Tb.Sp and Tb.N the worst (r = 0.53 and 0.6 respectively). However a significative statistical difference (*p* > 0.01) was reported for all the calculated features [13]. MRI images with an in plane pixel-size lower than the smallest trabecular thickness order (0.1 mm) are not easily reachable. On that basis, one cannot expect to fully characterize it. Moreover, these findings were further extended in a larger study conducted in 39 distal radius specimens scanned at 1.5T MRI (0.152 mm × 0.152 mm × 0.9 mm) and using contact radiography (0.05 mm isovolumetric). The results showed a significant correlation (r > 0.61) between bone microstructure parameters derived from both methods with Tb.Sp and BVF providing the highest correlations (r = 0.69 and *p* = 0.75 respectively) [102]. More recently, Rajakapse et al. conducted a study in 13 cylindrical specimens (7 proximal femurs, 3 proximal tibiae, and 3 third lumbar vertebrae) extracted from 7 human donors and computed microarchitectural parameters using 9.4T micro-MRI (0.050 mm isovolumetric) and μCT (0.021 mm isovolumetric). Architectural parameters were found to highly correlate between these two modalities with a slope close to unity (r^2^ ranging from 0.78 to 0.97) [116]. In a more recent study conducted in three cadaveric entire proximal femurs evaluating the trabecular morphology using 7T MRI (0.13 mm × 0.13 mm × 1.5 mm) and comparing the results with those acquired using μCT (0.051 mm isovolumetric) (Figure 1), Soldati et al. showed a good intraclass correlation coefficient for all the parameters (ICC > 0.54) between 7T and μCT [81] illustrating that bone morphological metrics of human specimens can be properly assessed using MRI. Moreover, due to the comparison between MR images and gold standard high-resolution CT images, it has been shown that trabecular features derived from images with a similar pixel size provide statistically comparable results. However, when assessing bone trabeculae using MRI, partial volume effects will occur and will affect image segmentation and trabeculae quantification.

#### 3.6.2. In-Vivo

The MRI potential for the bone microstructure has also been assessed in vivo in anatomical regions more affected by osteoporosis, i.e., tibiae and radii, vertebrae [65,117,118], distal [18,108,109,111,119], and proximal femurs [60,71,72]. Microarchitectural parameters extrapolated from 3T MRI (0.156 mm × 0.156 mm × 0.5 mm) and compared to HR-pQCT of tibiae and radii of 11 healthy volunteers showed good correlation for BVF (r = 0.83) and Tb.Sp (r = 0.7) in tibiae and good correlation for all the microarchitecture parameters investigated in radii (r = 0.65, 0.95, 0.83, and 0.63 for BVF, Tb.N, Tb.Sp, and Tb.Th respectively) [77]. Kazakia et al. extended these results in a study conducted in tibiae and radii of 52 postmenopausal scanned at 3T MRI (0.156 mm × 0.156 mm × 0.5 mm) and using HR-pQCT. A significant correlation between MRI and HR-pQCT has been reported for Tb.N (r^2^ = 0.52) and Tb.Sp (r^2^ = 0.54–0.60) with no statistical difference for these two parameters. Poor correlations were reported for BVF and Tb.Th (r^2^ = 0.18–0.34) [120]. Similar results were also reported by Folkesson et al., in a study conducted in 52 postmenopausal women scanned at 3T (0.156 mm × 0.156 mm × 0.5 mm) and using HR-pQCT in both tibiae and radii. All the structural parameters derived from MRI were highly correlated to those obtained from HR-pQCT (Tb.N was equal to 0.68 and 0.73 and Tb.Sp was equal to 0.77 and 0.67 for tibiae and radii respectively) with the exception of BVF and Tb.Th for which correlations were less significant (BVF was equal to 0.61 and 0.39 and Tb.Th was equal to 0.43 and 0.32 for tibiae and radii respectively) [113]. Furthermore, Krug et al. confirmed and extended these results in a study conducted in distal tibiae of 10 healthy volunteers scanned at 3T and 7T (0.156 mm × 0.156 mm × 0.5 mm for both techniques). The results showed that microarchitectural parameters extracted from HR-pQCT images had higher correlation with those extracted from 7T MR images (r equal to 0.73 for BVF, 0.69 for Tb.N, 0.89 for Tb.Sp, and 0.13 for Tb.Th) as compared to 3T MR images (r = 0.83, 0.49, 0.67, and 0.15 for BVF, Tb.N, Tb.Sp, and Tb.N respectively) (Figure 1). Interestingly, the corresponding absolute values did only differ by 0.6% for 7T and 3% for 3T [88]. All the findings reported above indicate good correlations for Tb.Sp and Tb.N between MRI and HR-pQCT. In contrast, this was not the case for BVF and Tb.Th. The limited resolution in MRI leads to partial volume effects responsible for the exclusion of the smallest trabeculae, while susceptibility artifacts enhance the remaining trabeculae leading to an overestimation of Tb.Th. This double effect seems limited when using UHF MRI. Indeed, good correlations were found between MRI and HR-pQCT metrics although a poor correlation was still existing for Tb.Th.

### 3.7. Reported Limitations

The main limitation reported regarding the bone morphology evaluation using MRI is related to partial volume effects resulting from the ratio between the resolution offered by the MR machines and the trabecular dimension. The minimum trabecular size is in the order of 0.1 mm. If the pixel size is larger than this limit, trabecular broadening would occur. In the worst possible scenario, trabecular broadening would cause the disappearance or the aggregation of the finest trabeculae leading to over- or underestimation of the main morphological characteristics [13,62,81,88,116]. Majumdar et al. reported an overestimation of the BVF (3 times) and the Tb.Th (3 times) and an underestimation of the Tb.Sp (1.6 times) in the MR images (0.156 mm × 0.156 mm × 0.3 mm) compared to the 18-µm µCT images [13]. Many studies conducted in different anatomical sites both in vivo and in vitro have shown that increasing the main magnetic field strength may emphasize small trabecular structures, normally not visible due to partial volume effects and susceptibility-induced broadening [81,88]. Moreover, several studies have shown that spin echo sequence are less prone to partial volume effects than gradient echo ones [77,88,100]. In particular, Krug et al. compared gradient echo and spin echo sequences at 7T and the results showed that SE sequence provided decreased BVF (−13%) and Tb.Th (−23%) and an increase in Tb.N (13%) and Tb.Sp (1%) as compared to gradient echo [88]. SE sequences have shown their higher discriminative power to resolve the bone microstructure due to a more homogeneous bone marrow signal. However, at UHF spin-echo sequences should be used carefully due to a specific absorption rate (SAR) that limits the number of acquirable images. Soldati et al. reported a maximum number of 10 acquired images using a turbo spin echo sequence at 7T in approximately 16 min.

MR imaging conducted in cadaveric specimens may suffer from an additional limitation related to air bubbles trapped in the trabecular network and leading to magnetic susceptibility effects [81,116]. Air bubbles provide grey level intensities similar to the bone signal so that pixel misclassification could be expected. In order to properly perform MRI acquisition of cadaveric specimens air bubbles have to be removed using different strategies that have been reported and validated through images analysis [81,107,116]. The common strategy used mainly for small specimens (<5 cm^3^) is related to the bone marrow removal through a gentle water jet, the immersion in 1 mM Gd-DTPA saline solution to simulate the bone marrow magnetic response and the removal of air bubbles using centrifugation (approximately between 5× to 6× *g*, for 5 min) [13,107,116]. Hipp et al. reported an alternative solution consisting in filling marrow spaces with confectioners’ sugar to provide contrast between bone and marrow [78]. More recently, Soldati et al. reported no trabecular misclassification due to air bubbles by combining vacuum application and vibrational forces to large cadaveric specimens (entire proximal femurs) immersed in 1 mM Gd-DTPA saline solution [81].

## 4. Prospectives

In this chapter we provide an overview of the most recent results reported in the literature, which are related to the assessment of bone marrow using magnetic resonance spectroscopy and chemical shift encoding MRI, bone phosphorus content, and bone mineral density using solid-state MRI and quantitative susceptibility measurements. These techniques are considered promising to further investigate bone quality.

### 4.1. Magnetic Resnance Spectroscopy vs. Chemical Shift Encoding-MRI

Several MRI studies have shown that bone marrow, which is mainly composed by adipocytes (yellow marrow regions) and hematopoietic red blood cells (red marrow regions), may play a key role in the bone health and metabolism. Moreover, it has been reported that distinct alterations become increasingly evident when comparing osteoporotic subjects to controls [121,122]. The bone marrow fat content can be assessed from bone marrow fat fraction (BMFF) and proton density fat fraction (PDFF) measurements [122,123]. Bone marrow has been actually investigated using magnetic resonance spectroscopy (MRS) and chemical shift encoding based water fat MRI (CSE-MRI).

Up to now, the most frequently used technique for bone marrow quantification has been the single-voxel proton MRS, which is also considered the gold-standard. Based on a localized scheme, water and fat components can be quantified on the basis of their respective resonance frequencies. Point-resolved spectroscopy (PRESS) and stimulated echo acquisition mode (STEAM) have been mainly used. Given that the STEAM sequence allows shorter TEs as compared to PRESS, a higher signal can be expected for the short-T2 water component of the BM spectrum [122].

MRS has been used to assess BMFF at the spine level [31,33,37,124,125,126,127] and fewer studies have been devoted to the proximal femur [128,129,130]. Correlations between BMFF and BMD or T-scores have been repeatedly reported. BMFF is elevated in osteoporotic patients and negative correlations have been reported between BMFF and BMD. He et al., in a study conducted in L2–L4 vertebrae of 123 subjects (49 with normal bone density, 38 with osteopenia and 36 with osteoporosis) scanned using PRESS at 3T (voxel size 15 × 15 × 15 mm^3^) showed that BMFF was increased in patients with reduced BMD values while the metrics were negatively correlated (r = −0.82, *p* < 0.001) [125], further confirming previous results on the lumbar spine (L1-L2) and proximal femurs [131]. In a study conducted in femoral neck of 33 postmenopausal woman using 3T MRS (PRESS sequence, single voxel 15 × 15 × 15 mm^3^), Di Pietro et al. reported larger content of methylene (L13), glycerol (L41,L43), and total lipid in osteoporotic subjects [129]. There changes suggest that MRS of bone marrow lipid profiles from peripheral skeletal sites might be a promising screening tool to identify individuals with or at risk of developing osteoporosis [129,132].

CSE-MRI can be used to obtain a spatially resolved quantification of BMFF. Multi-echo GRE sequences are commonly used with an appropriate selection of experimental parameters (i.e., small flip angle to reduce T1 bias, and distinct correction of T2 * decay effects during the postprocessing stage) [122,126,133,134]. T2 * decay effects have to be particularly considered when measuring the PDFF. In fact, T2 * of trabecular bone is reduced due to microscopic magnetic field inhomogeneity effects [122].

A good agreement has been reported between BMFF measured using MRS and CSE-MRI for both spine and proximal femur [135,136,137]. At the spine level, BMFF has been reported as increased in osteoporotic patients and inversely correlated with BMD and T-scores [122,135,138]. At the proximal femur level, Martel et al., reported a higher saturation (+14.7% to +43.3%), and a lower mono- (−11.4% to −33%) and polyunsaturation (−52% to −83%) in postmenopausal women. More specifically, red marrow of postmenopausal women showed a lower fat content (−16% to −24%) and a decreased polyunsaturation (−80% to −120%) in the femoral neck, greater trochanter, and Ward’s triangle [139]. In another study, it has been reported that PDFF derived from CSE-MRI would discriminate benign osteoporotic and malignant vertebral fractures. Accordingly, Schmeel et al. reported that both PDFF and PDFF_ratio_ (fracture PDFF/normal vertebrae PDFF) of malignant VCFs were significantly lower as compared to acute benign (PDFF, 3.48 ± 3.30% vs. 23.99 ± 11.86% (*p* < 0.001) and PDFF_ratio_, 0.09 ± 0.09 vs. 0.49 ± 0.24 (*p* < 0.001)). The corresponding areas under the curve were 0.98 and 0.97 for PDFF and PDFF_ratio_ respectively providing a 96% and 95% accuracy for the discrimination between acute benign and malignant VCFs [140]. CSE-MRI conducted in 156 subjects at 3T (8 echoes 3D spoiled gradient echo sequence, voxel size 0.98 mm × 0.98 mm × 4.00 mm, acq. time 1:17 min), showed that vertebral bone marrow heterogeneity is primarily dependent on sex and age but not on anatomical location suggesting that future studies should investigate the bone marrow heterogeneity with regards to aging and disease [141]. Baum et al. in a study conducted on the whole spine of 28 young and healthy patients using CSE-MRI at 3T (8 echoes, acq. time 3:15 min), extend these results reporting that the repeatability of PDFF measurements expressed an averaged absolute precision error of 1.7% over C3-L5 [142].

MRS and CSE-MRI have enabled the evaluation of the nonmineralized bone compartment and the extraction of the PDFF. The marrow adipose tissue has shown to have a role in bone health, through its paracrine and endocrine interaction with the other bone components. However, the implication of marrow adipose tissue in physiological and pathological conditions remains unclear [143].

### 4.2. MR Susceptibility

Magnetic susceptibility is the macroscopic physical quantity that describes the tissue’s induced magnetization in the presence of an external magnetic field. Since the early days of MRI, susceptibility quantification has been considered as of paramount interest given that it is related to the tissue’s chemical composition. Even a small susceptibility change can lead to field distortions that could be quantified. This has been achieved through SWI (susceptibility weighted imaging) [144,145] and quantitative aspects could be computed from MRI phase and magnitude signals by means of QSM (quantitative susceptibility mapping) [146,147].

Dense calcified tissues, such as bone, show a strong diamagnetic behavior. On that basis, QSM [148] could be used to assess bone mineral changes [149]. Although QSM has been largely developed for brain imaging [150,151], the corresponding applications for bone are still considered very challenging. Cortical bone has a very low signal using conventional echo times GRE imaging and water connected to the crystalline mineral structures or to the collagen matrix has an ultrashort transverse relaxation time (T2 * = 300 μs [152]) thereby showing a non-meaningful signal for QSM. In order to overcome this issue, ultrashort echo-time (UTE) GRE imaging [148] has been developed to obtain phase information for reliable QSM, which may be used in the evaluation of BMD [153].

For example, correlations between QSM and BMD have been studied through clinical MRI sequences in spine and ankle trabecular bones [153,154]. In Chen et al. [153], the efficacy of QSM in the assessment of osteoporosis for post-menopausal women was investigated. The L3 vertebrae body of 70 post-menopausal women was studied through a multi-GRE UTE sequence on a 3T MR system (TR = 20 ms, TE = 0.142, 2.4, 4.6, and 6.8 ms, voxel size = 1.0 mm × 1.0 mm × 2.0 mm, and acq. time = 10 min) and a qCT examination. Based on qCT values, individuals were divided into normal and affected by osteopenia or osteoporosis. The QSM values were higher for the osteoporosis group than in either the normal or the osteopenia group (*p* < 0.001) and showed an highly negative correlation with qCT values (r = −0.72, *p* < 0.001) [153].

Non-UTE multi-GRE sequences can be applied to QSM: ankle in vivo imaging was performed by Diefenbach et al. using a time-interleaved gradient-echo sequence (TIM-GRE) at 3T (9 echoes, TE_min_ = 1.25 ms with ΔTE = 0.7 ms, voxel size = 1.5 mm × 1.5 mm × 1.5 mm, and acq. time = 7 min) in order to evaluate the feasibility of QSM for trabecular bone imaging and investigate its sensitivity for measuring trabecular bone density [154]. Mean susceptibility values in calcaneus regions with different trabecular bone density were compared to CT attenuation values. In highly trabecularized regions, qCT values showed significant correlation with lower susceptibility values (r = −0.8, *p* = 0.001) [154]. In addition, differences in calcaneus trabecularization were outlined in QSM maps in good agreement with qCT and high resolution MR images.

Furthermore, cones 3D UTE-MRI techniques have recently been developed showing similar susceptibility values but faster scanning process if compared with other different sampling strategies [155]. In Jerban et al. [156], cones 3D UTE-MRI was implemented for ex vivo QSM in order to investigate correlations of susceptibility with volumetric intracortical BMD in human tibial cortical bone. Nine tibial midshaft cortical bones specimens were scanned in a 3T clinical scanner with a cones 3D UTE-MRI sequence for QSM (TE = 0.032, 0.2, 0.4, 1.2, 1.8, and 2.4 ms, voxel size = 0.5 mm × 0.5 mm × 2.0 mm, and acq. time = 20 min) and with high-resolution µCT for BMD estimation (Figure 2). Average QSM values were calculated in one slice (2 mm thickness) at the middle of the specimen and showed a strong correlation with volumetric BMD (r = −0.82, *p* = 0.01) and bone porosity (r = 0.72, *p* = 0.01). Results in this study highlight the potential of 3D cones UTE-MRI QSM as a possible future tool in the in vivo diagnosis of bone diseases that can be detected through mineral level variations in cortical bone.

Despite its potential in providing an X-ray radiation free approach to quantify susceptibility in bone tissue, QSM suffers from some limitations. Data processing is relatively complex and still under study while data acquisition times are too long if compared to clinical MRI sequences. Furthermore, bone susceptibility variations due to soft surrounding tissues should be taken into account in future in vivo clinical studies and applications.

### 4.3. Solid State MRI

Solid state MRI has been recently described in a review by Seifert and Wehrli [157]. One of the main issued faced by MRI of the solid part of bones is the extremely weak MR signal. In order to acquire the fast decaying (i.e., short T2 *) ^1^H and ^31^P MRI signals in bone, the time between signal excitation, encoding, and acquisition must be shorter than the one used in conventional MRI sequences [157,158]. Hence, to image short-T2 * tissues three solid state radial pulse sequences have emerged: ultrashort echo time (UTE) [159], zero echo time (ZTE) [160,161], and sweep imaging with Fourier transformation (SWIFT) [162,163,164]. The main strategy to image short-T2 * tissue is to reduce the time delay between the end of the signal excitation and the beginning of encoding and acquisition. In UTE, the time delay is reduced by beginning the signal encoding and acquisition simultaneously and immediately after the MRI system’s transmit/receive switching dead time has elapsed [159,165]. In ZTE, signal encoding is begun simultaneously with the excitation, but the time delay to signal acquisition is still dictated by the transmit/receive dead time resulting in the loss of the first data points in the acquisition [166,167,168]. In SWIFT, the three steps (excitation, encoding, and acquisition) are performed in a finely interleaved (gapped) [162] of fully simultaneous (continuous) [163] manner, allowing the in vivo imaging of teeth, where the T2 * is even shorter than that of bone [169]. All these sequences have been used ex vivo and in vivo applying whole-body MRI scanners at different field strengths. However, in the case of in vivo solid-state NMR of tibial shafts, a minimum voxel size of 0.98 mm^3^ for ^1^H and 2.5 mm^3^ for ^31^P is required to have enough SNR at an acceptable acquisition time (<10 min for ^1^H and <25 min for ^31^P) and to avoid SAR limitation [157,170].

Solid-state MRI could be used to compute total bone water (TW), water bound to the collagen matrix (BW), and pore water (PW). Several consistent studies from different groups have been reported for bone extremities [63,170,171,172,173] showing also the ability to differentiate between pre- and post-menopausal women. Techawiboonwong et al., in a study conducted in distal tibiae of pre-menopausal and post-menopausal women (*n* = 5 for each group) scanned using an UTE sequence at 3T MRI (pixel size 0.3 mm × 0.3 mm × 8.0 mm and acq. time = 9 min) reported a difference in the TW concentration of 17.4% and 28.7% respectively for the pre- and post-menopausal groups [171]. Moreover, BW and PW were acquired in vivo tibiae and wrist of 5 volunteers using 3T MR scanner (isotropic pixel size of 1.5 mm in the leg and 1.2 in the wrist and acq. time = 8–14 min per acquisition) reporting a mean BW of 34.86 ± 2.59 M and a mean PW 6.14 ± 1.97 M, similar to previously ex vivo observations [172,173,174].

Using ^31^P NMR, solid-state MRI could also be used to quantitatively assess the mass of bone mineral in bone tissue [175]. An ex vivo study conducted in 16 tibiae specimens acquired using ZTE ^31^P at 7T (pixel size = 3.84 mm isovolumetric and acq. time = 3 h and 3 min) and ^1^H at 3T (pixel size = 1.17 mm isovolumetric and acq. time = 26 min and 45 s) by Seifert et al. (Figure 2), reported a mean bone mineral ^31^P density of 6.74 ± 1.22 M and mean BW ^1^H density of 31.3 ± 4.2 M [14]. In addition, ^31^P and BW densities correlated positively with pQCT density (^31^P: r^2^ = 0.46, *p* < 0.05; BW: r^2^ = 0.50, and *p* < 0.005), showing that MRI-based measurements are able to detect intersubject variations in apparent mineral and osteoid density in human cortical bone using clinical hardware [14,176]. However, Tamimi et al. in a study conducted on trabecular femur head samples collected from patients who had hip fractures and individuals with osteoarthritis reported no differences in neither ^1^H nor ^31^P between the two groups [177].

In a more recent study performed on in vivo tibiae of 10 healthy subjects, Zhao et al. acquired at 3T ^1^H UTE (pixel size = 0.98 mm^3^ and acq. time = 8:20 min) and ^31^P ZTE (pixel size = 2.5 mm^3^ and acq. time = 22:30 min). They showed no differences in the ^31^P concentration in healthy adults across 50-years of age [178] and a strong positive correlation (r = 0.98, *p* < 0.0001) between bone mineral content (BMC) measured using ^31^P MRI and HR-pQCT [170], further extending previous in vivo studies [179].

Therefore, solid-state MRI has shown its potential as translational techniques into clinical research and practice providing information related to the mineral composition of bone tissue, bound, and pore water. However spatial resolution, SNR, and scan time remain key challenges for the solid state MRI [157,170,180]. These three characteristics are dependent on each other and a trade-off has to be established to retrieve useful information in a clinically acceptable acquisition time. Usually, a SNR higher or equal to 10 is recommended and, to maintain the acquisition time in an acceptable range, the voxel size is enlarged along the bone axis where features are considered to be constant.

## 5. Conclusions

Over the last decades, the multiple technical improvements that have been made in MRI have opened new MRI applications such as a bone microarchitecture assessment. Up to now, most of the MRI studies conducted in bones have been performed using 1.5T and 3T scanners. However, results obtained at UHF showed the technical advantages and the higher discriminative power of 7T MRI for the assessment of the bone microstructure of the most proximal anatomic locations, including those more affected by osteoporotic fractures. The advantages provided by UHF MRI have shown great potential on the bone microstructure assessment and made this technique almost ready for a daily clinical application.

Moreover, bone morphological parameters derived from both specimens and patients acquired using MRI were shown to provide features in the same range of those derived with the gold-standard X-rays techniques, with the great advantage of being completely non-invasive for the patients. In addition, MRI was also shown to be able to provide supplementary information about the mineral content, i.e., phosphorous density, not accessible using X-rays techniques. Furthermore, MRI microarchitecture analysis was able to evaluate changes related to age and/or pathology suggesting the great clinical potential for MRI in evaluating different bone pathologies, assessing the risk stratification, and following the therapy delivery.

Up to now, BMD derived from DXA measurements was the only parameter used to identify bone related pathologies. Several studies have demonstrated that microarchitectural parameters provide additional information regarding the skeletal fragility and should be integrated with BMD to provide a more comprehensive view of the bone quality. MRI is completely radiation free and the application of UHF MRI made accessible the anatomical regions further away from the skin surface, with a resolution in the same range of trabecular thickness, and in an acquisition time compatible for in vivo clinical use. Moreover, MRI, and in particular UHF MRI, showed to provide bone morphological parameters in the same range of gold standard analysis both in specimens and patients.

Finally, MSC and CSE-MRI, solid state MRI, and QSM have shown to be useable in in vivo acquisitions providing bone marrow fat quantification, mineral composition of bone tissue, bound, and pore water, and magnetic susceptibility quantification. However, for the clinical application of solid state MRI and QSM acquisition times would have to be reduced. MRI could certainly be added to BMD measurements for a complete analysis of bone quality, health, and metabolism.

## Figures and Tables

**Figure 1 ijms-22-02509-f001:**
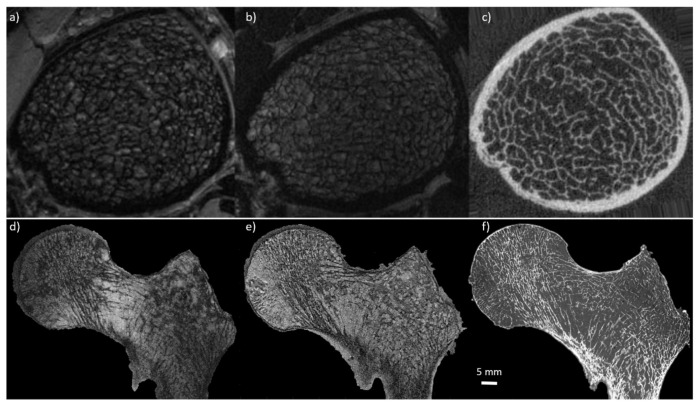
Comparison between MRI and CT. (first row) MR images of in vivo distal tibia acquired using gradient echo sequence at 7T MRI (**a**) (0.156 mm × 0.156 mm × 0.5 mm) and 3T MRI (**b**) (0.156 mm × 0.156 mm × 0.5 mm), and compared with high-resolution peripheral computed tomography (HR-pQCT) (**c**) (0.082 mm^3^) (reproduced from J. of Mag. Res. Im. 27:854–859 (2008)). (second row) MR images of cadaveric proximal femur acquired using turbo spin echo sequence at 7T MRI (**d**) (0.13 mm × 0.13 mm × 1.5 mm) and 3T MRI (**e**) (0.21 mm × 0.21 mm × 1.1 mm), and compared with µCT (**f**) (0.051 mm^3^). Note that using MRI, the trabecular bone appears black and bone marrow delivers the bright signal whereas for HR-pQCT and µCT the trabecular bone is shown bright. Additionally, note that the trabecular network is clearly more enhanced at 7T compared to 3T.

**Figure 2 ijms-22-02509-f002:**
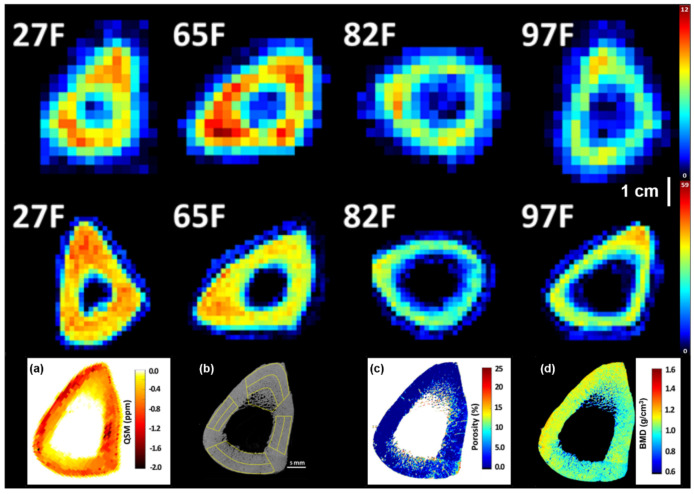
Solid state MRI and quantitative susceptibility mapping. (first row) Maps of bone mineral ^31^P density, and bound water density (second row) in central slices of 4 human tibial cortical bone specimens. Age and gender of bone specimen donors are indicated. Bone mineral ^31^P and bound water ^1^H densities are markedly lower in bones from elderly female donors than from younger females or males. ^31^P maps also suffer from increased point spread function blurring because of the lower gyromagnetic ratio and shorter T2 * of ^31^P. (reproduced from NMR Biomed. 27: 739–748 (2014)) (third row) (**a**) QSM map obtained through Cones 3D UTE-MRI scans (0.5 mm × 0.5 mm × 2 mm voxel size) of a tibial midshaft cortical bone (45-year-old female), (**b**) one µCT slice at 9 µm isotropic voxel size, (**c**) µCT-based porosity, and (**d**) BMD map of the same specimen. Local maxima in the QSM map correspond to high BMD regions and low porosity values in µCT-based maps (reproduced from Magn. Res. Im. 62: 104–110 (2019)).

**Table 1 ijms-22-02509-t001:** List of the main magnetic resonance imaging (MRI) parameters and sequences.

Anatomical Site	Clinical History	Specimen /Patient	Acq. Time	Sl. Thickness [mm] [mm]	Pix. Size [mm]	FOV [mm]	Sequence	Main Field	N°	Reference
distal radii	type 2 diabetes	patient	12 min 9 s	1	0.195 × 0.195	100 × 100	FSE	1T	[78]	Pritchard et al.
calcaneus	osteoporotic hip fractures	patient	15 min 15 s	0.5	0.195 × 0.195	100 × 100	GE	1.5T	[67]	Link et al.
distal radii	healthy	patient	16 min 25 s	0.5	0.156 × 0.156	80 × 45	3D FLASE	1.5T	[75]	Techawiboonwong et al.
distal radii	healthy	patient	3 min 15 s	0.5	0.156 × 0.156	80 × 45	3D SSFP	1.5T	[75]	Techawiboonwong et al.
distal radii	NA	specimen	15 min	0.3	0.156 × 0.156	80	GE	1.5T	[13]	Majumdar et al.
lumbar spine	osteoporotic	patient	16 min	0.7	0.156 × 0.156	80 × 80	GE	1.5T	[65]	Majumdar et al.
distal radii	hip fractures	patient	NA	0.5	0.156 × 0.156	80 × 80	GE	1.5T	[16]	Majumdar et al.
distal radii	NA	specimen	58 min (1) 16 min (2)	0.3 (1) 0.9 (2)	0.153 × 0.153	49×78	SE	1.5T	[79]	Link et al.
prox. femur	NA	specimen	74 min (1) 27 min (2)	0.3 (1) 0.9 (2)	0.195 × 0.195	75 × 100	SE	1.5T	[80]	Link et al.
prox. femur	healthy	patient	6 min 12 s	1.5	0.234 × 0.234	NA	3D FIESTA	1.5T	[71]	Krug et al.
distal tibiae	NA	specimen	40 min	0.16	0.160 × 0.160	70 × 63	3D FLASE	1.5T	[81]	Rajapakse et al.
lumbar spine	NA	specimen	15 min 23 s	0.41	0.137 × 0.137	70 × 64 × 13	3D FLASE	1.5T	[70]	Rajapakse et al.
distal radii(1) distal tibiae(2)	osteopenic and osteoporotic	patient	12 min (1) 16 min (2)	0.4	0.137 × 0.137	70 × 40(1) 70 × 50(2)	3D FLASE	1.5T	[66]	Ladinsky et al.
distal femur	cerebral palsy (children)	patient	9 min 52 s	0.7	0.175 × 0.175	90	3D fast GE	1.5T	[82]	Modlesky et al.
distal radii(1) distal tibi.ae(2)	osteoporotic	patient	12 min (1) 16 min (2)	0.41	0.137 × 0.137	70 × 40 × 13 (1) 70 × 50 × 13 (2)	3D FLASE	1.5T	[83]	Rajapakse et al.
prox. femur	NA	specimen	16 min 55 s	1.1	0.21 × 0.21	120	TSE	3T	[84]	Soldati et al.
prox. femur	healthy	patient	12 min 43 s	1.5	0.234 × 0.235	NA	3D FIESTA	3T	[71]	Krug et al.
distal radii, distal tibiae	NA	specimen	<10 min	0.5	0.156 × 0.156	NA	GE	3T	[77]	Krug et al.
distal radii, distal tibiae	NA	specimen	<10 min	0.5	0.156 × 0.156	NA	GRE	3T	[77]	Krug et al.
distal radii, distal tibiae	NA	specimen	<10 min	0.5	0.156 × 0.156	NA	SE	3T	[77]	Krug et al.
distal tibiae	osteoporotic	patient	15 min	0.41	0.137 × 0.137	70 × 64 × 13	3D FLASE	3T	[69]	Zhang et al.
prox. femur	fragility fractured	patient	25 min 30 s	1.5	0.234 × 0.234	120	FLASH	3T	[60]	Chang et al.
prox. femur	long-term glucocorticoid	patient	15 min 18 s	1.5	0.234 × 0.234	100	FLASH	3T	[72]	Chang et al.
distal radii	HR+ breast cancer	patient	7 min	0.34	0.170 × 0.170	65	GE	3T	[85]	Baum et al.
distal femur	osteoarthritis	patient	9 min 18 s	1	0.180 × 0.180	100	3D B-FFE	3T	[86]	Liu et al.
prox. tibia	osteoarthritis	patient	3 min	2.8	0.230 × 0.240	120 × 123	SE	3T	[87]	MacKey et al.
prox. tibia, distal femur	osteoarthritis	patient	NA	1	0.195 × 0.195	100	FIESTA-c	3T	[88]	Chiba et al.
prox. tibia, distal femur	osteoarthritis	patient	NA	1	0.195 × 0.195	160	SPGR	3T	[88]	Chiba et al.
distal tibiae	NA	specimen	7 min	0.41	0.137 × 0.137	70 × 53 × 13	3D FLASE	3T	[19]	Rajapakse et al.
prox. femur	NA	specimen	16 min 45 s	1.5	0.13 × 0.13	130	TSE	7T	[89]	Soldati et al.
prox. femur	NA	specimen	37 min 36 s	0.5	0.170 × 0.170	140 × 140	GRE	7T	[62]	Guenoun et al.
distal tibiae	healthy	patient	19 min 10 s	0.5	0.156 × 0.156	NA	SE	7T	[17]	Krug et al.
distal tibiae	healthy	patient	18 min 25 s	0.5	0.156 × 0.157	NA	FP	7T	[17]	Krug et al.
vertebrae (1 axial, 2 sagittal)	NA	specimen	34 min (1) 51 min (2)	0.4 (1) 0.5 (2)	0.170 × 0.170	140 × 140	GRE	7T	[90]	Guenoun et al.
distal femur	fragility fractured	patient	7 min 9 s	1	0.234 × 0.234	120	FLASH	7T	[18]	Chang et al.
femurs, tibiae, vertebrae	NA	specimen	120 min	0.05	0.05 × 0.05	6.4 × 6.4 × 25.6	SE	9.4T	[91]	Rajapakse et al.

## Data Availability

No new data were created or analyzed in this study. Data sharing is not applicable to this article.

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
