# Peer review of "Survey of MRI Usefulness for the Clinical Assessment of Bone Microstructure"

_ijms, 2021, doi:10.3390/ijms22052509_

Round 1

Reviewer 1 Report

While considerable work seems to have gone into this review, it does not seem to reflect state of the art as far as MR goes. There is some focus on anatomical MRI but even that is not up to date. The following suggestions are recommended:

Lines 103-121: Reference 5 is cited several times to talk about the global burden of MSK disorders – yet it is a paper from 2006. Please update your references and cite more recent figures. This is a pattern throughout the manuscript – a lot of references are outdated and need to be replaced with state of the art references.

Lines 227-228: Please specify what kind of MR images (T1-weighted, T2-weighted, etc.) – there is no such thing as conventional MRI images.

Lines 286-294: This paragraph deals with 7T imaging. Since this manuscript mainly deals with imaging the bone micro-architecture in a clinical setting, what is the availability of 7T scanners in clinical settings? 7T scanners are pretty much limited to research settings currently.

Lines 296-308: Once again, a lot of the literature cited here is very old and considerable work has been done since then. For example, there is considerable recent literature looking at correlations between BMD and BMAT using MR spectroscopy and other methods.

This is a review article and needs to be much more up to date and needs to reflect the current state of the field. While the authors have put in a considerable amount of work on this manuscript, a comprehensive literature review and update of all references is needed.

Reviewer 2 Report

In this review article, the authors presented a survey regarding the useful information MRI could provide for the assessment of the inner trabecular morphology of different bone segments. They also discussed upcoming MRI applications (solid-state MRI and quantitative susceptibility mapping) which could provide additional biomarkers for the assessment of bone microarchitecture.

Even though this review is very important in the field of MRI, but this manuscript needs more modification and correction before publication. e.g. language used in the paper is very poor.

Author Response

Answers to Reviewer 2:

Point1: In this review article, the authors presented a survey regarding the useful information MRI could provide for the assessment of the inner trabecular morphology of different bone segments. They also discussed upcoming MRI applications (solid-state MRI and quantitative susceptibility mapping) which could provide additional biomarkers for the assessment of bone microarchitecture.

Even though this review is very important in the field of MRI, but this manuscript needs more modification and correction before publication. e.g. language used in the paper is very poor.

Answer to Reviewer 2:

According to the reviewer’s suggestions, the manuscript has been reviewed by a native English colleague.

Round 2

Reviewer 1 Report

The authors have updated the reference list and added a section on MRS, which is the area where the cutting edge research into marrow fat and bone strength is happening. This makes it a stronger manuscript.